# The Role of Non-Coding RNAs in Glioma

**DOI:** 10.3390/biomedicines10082031

**Published:** 2022-08-20

**Authors:** Anshika Goenka, Deanna Marie Tiek, Xiao Song, Rebeca Piatniczka Iglesia, Minghui Lu, Bo Hu, Shi-Yuan Cheng

**Affiliations:** 1The Ken & Ruth Davee Department of Neurology, Lou & Jean Malnati Brain Tumor Institute at Northwestern Medicine, The Robert H. Lurie Comprehensive Cancer Center, Northwestern University Feinberg School of Medicine, Chicago, IL 60611, USA; 2Master of Biotechnology Program, Northwestern University, Evanston, IL 60208, USA

**Keywords:** non-coding RNA, glioma, microRNA, long non-coding RNA, circular RNA

## Abstract

For decades, research in cancer biology has been focused on the protein-coding fraction of the human genome. However, with the discovery of non-coding RNAs (ncRNAs), it has become known that these entities not only function in numerous fundamental life processes such as growth, differentiation, and development, but also play critical roles in a wide spectrum of human diseases, including cancer. Dysregulated ncRNA expression is found to affect cancer initiation, progression, and therapy resistance, through transcriptional, post-transcriptional, or epigenetic processes in the cell. In this review, we focus on the recent development and advances in ncRNA biology that are pertinent to their role in glioma tumorigenesis and therapy response. Gliomas are common, and are the most aggressive type of primary tumors, which account for ~30% of central nervous system (CNS) tumors. Of these, glioblastoma (GBM), which are grade IV tumors, are the most lethal brain tumors. Only 5% of GBM patients survive beyond five years upon diagnosis. Hence, a deeper understanding of the cellular non-coding transcriptome might help identify biomarkers and therapeutic agents for a better treatment of glioma. Here, we delve into the functional roles of microRNA (miRNA), long non-coding RNA (lncRNA), and circular RNA (circRNA) in glioma tumorigenesis, discuss the function of their extracellular counterparts, and highlight their potential as biomarkers and therapeutic agents in glioma.

## 1. Introduction

RNA was once thought of as just a messenger molecule for relaying the code from DNA by ribosomes to synthesize proteins. However, in the past three decades, different types of RNA molecules have been discovered, the most important of which are the non-coding RNAs (ncRNAs). ncRNAs account for 90% of the RNA species transcribed from the human genome [1]. The discovery of tens of thousands of ncRNAs has augmented their importance in physiology and disease [1,2,3]. For decades, it was believed that only 2% of the human genome that codes for proteins was functional and the remaining 98% was non-functional, or ‘junk’ [4,5]. Only after the ENCODE project was completed, did it become appreciable that the non-protein coding region of the genome is transcribed into thousands of RNA molecules, which are not only involved in numerous fundamental life processes such as growth, differentiation, and developmental processes, but also play critical roles in a wide spectrum of human diseases, including cancer [6,7,8].

The discovery of ncRNAs has added a new dimension to the diagnosis and treatment of cancer. Dysregulated ncRNA expression is found to affect cancer initiation, progression, and therapy resistance, and occurs through transcriptional, post-transcriptional, or epigenetic processes in the cell [1,9]. Genetic alterations in genes to code for ncRNAs are also associated with cancer. For example, deletion of 13q14.3 in chronic lymphocytic leukemia (CLL) results in deletion of miR-15/16 tumor suppressors [10]. Conversely, amplification of chromosomal regions has led to an increase of oncogenic long non-coding RNAs (lncRNAs) *FAL1* and *PVT1* [11,12]. Moreover, recurrent driver mutations in the promoters of lncRNAs *NEAT1* and *RMRP* can lead to altered expression and breast cancer progression [13].

Gliomas are common, and are the most aggressive type of primary tumors, which account for ~30% of central nervous system (CNS) tumors. Gliomas are constituted of pilocytic astrocytoma, oligodendroglioma, astrocytoma, oligoastrocytoma, ependymoma, and glioblastoma, which are classified by the World Health Organization (WHO) into grade I to IV tumors [14]. Despite the standard of care including surgical resection, radiotherapy, and chemotherapy, the prognosis of glioma patients remains poor [15]. Of these, glioblastoma (GBM), which are grade IV tumors, are common, and are the most lethal brain tumors. Only 5% of GBM patients survive beyond 5 years upon diagnosis [16]. The current treatment regimen had only increased the median survival to 14.6 months in adults [17,18]. Though recent clinical trials have shown that the addition of Tumor-treating fields (TTFields) increased the overall median survival by 4.9 months [19]. Another Grade IV astrocytoma, called diffuse midline glioma is a rare type of glioma and shows poor prognosis [20]. Thus, a deeper understanding of the cellular non-coding transcriptome might help identify biomarkers and therapeutic agents for a better treatment of glioma.

NcRNAs had been broadly classified into housekeeping RNAs and regulatory RNAs. Housekeeping RNAs include transfer RNA (tRNA), ribosomal RNA (rRNA), small nuclear RNA (snRNA), and small nucleolar RNA (snoRNA), which are involved in maintaining the regular proteome components of the cell. All other ncRNAs are involved in regulating gene expression in the cell and are categorized based on their size. The small ncRNA include microRNA (miRNA), PIWI-interacting RNAs (piRNAs), and tRNA-derived small RNAs (tsRNAs), while those greater than 200 nucleotides (nts) in length include lncRNAs, which include circular non-coding RNAs (circRNAs) and pseudogenes [21].

This review focuses on the recent developments and advances in ncRNA biology pertinent to their role in glioma tumorigenesis and therapy response. Since studies of tsRNA in glioma are scarce, here, we delve into the functional roles of miRNA, lncRNA, circRNAs, and briefly, piRNA, in glioma development, progression, and therapy resistance, thus highlighting the potential and clinical implications of the ncRNAs in glioma (Figure 1).

## 2. MicroRNAs: Biogenesis and Function

miRNAs are natural RNA interference (RNAi) molecules produced in cells. They are transcribed in the cell from the nuclear genome (pri-miRNA), processed, and exported out of the nucleus as stem loop precursor pre-miRNAs. In the cytoplasm, these pre-miRNAs are trimmed to form duplex miRNAs, which are loaded on to Argonaut proteins, either individual strand or both, to form the RNA-induced silencing complex (RISC). The miRNA-guided RISC binds to target mRNAs to regulate their expression majorly by mRNA destabilization (~80%) or by repressing translation (~20%) [22]. miRNA-RISC complexes also function in the nucleus and mitochondria [23,24]. These RNAi agents are special because a single miRNA can regulate the expression of multiple genes involved in different cellular functions, hence, these serve as attractive candidates for biomarkers and therapeutic targets in diseases, including cancer [25].

### 2.1. miRNA Deregulation in Glioma

The role of miRNAs in gliomagenesis has been extensively studied over the past several years, the functions of miRNAs being described as oncogenes or tumor suppressors in glioma.

The diverse action of a single miRNA on multiple targets is well represented in glioma tumorigenesis. For example, enhancing the expression of the tumor suppressor miRNA miR-128-3p or miR-145-5p in GBM results in a plethora of anti-tumorigenic phenotypes, such as increase in apoptosis or senescence, repression in invasive, metastatic, or angiogenesis potential, and alleviation of drug resistance in GBM by regulation of multiple targets by miRNAs [26,27,28,29]. Inhibiting the expression of oncogenic miRNAs, such as miR-21-3p or miR-21-5p, can lead to tumor regression in cancer cells or even lead to tumor eradication, as in the case of miR-10b in GBM [30,31,32]. Furthermore, primary glioma cells stably expressing the miR-302-367 repressed the stemness, proliferation, and tumorigenicity of neighboring GBM cells in a paracrine manner [33]. MiRNA-31 encoded by the *MIR31HG* gene, positioned adjacent to *CDKN2A/B* at Chromosome 9, is compromised in >72% of GBM, and its reintroduction reduces tumor burden by inhibiting NF-κB signaling, independent of *CDKN2A/B* status [34]. An extensive list of GBM-associated miRNAs can be found at Shea at al [35] and a list of the recent miRNA studies in GBM is provided in Table 1.

miRNAs interact with various cancer-associated signaling pathways in GBM [36,37]. In *PTEN*-deficient GBM, miR-29a promoted growth and invasion, by activating Akt, repressing Sox4 transcription factor, and upregulating the invasion-promoting protein, HIC5 [38]. miR-34a shows tumor suppressive roles in GBM by targeting c-Met and Notch signaling pathways [39]. Also, miR-34a could modulate EGFR levels. In one study, by targeting the Yin Yang-1 (YY1) transcription factor, which stimulated EGFR levels, miR-34a deletion and *EGFR* amplification were associated with decreased survival in GBM patients [40]. The EGFR pathway in GBM is regulated by another miRNA, miR-7, which is a major regulator of cancer pathways, and its forced expression decreases viability and invasiveness in primary GBM cells [41]. Moreover, miRNA expression regulates the heterogeneity in GBM. miR-125b/miR-20b are expressed only in the proneural (PN) subtype of GBM and not the mesenchymal subtype, and they regulate the Wnt signaling in the PN subtype by repressing the transmembrane protein FDZ6, thus contributing to the heterogeneity in GBM [42].

### 2.2. miRNA and Blood-Brain Barrier

The blood-brain barrier (BBB) is a highly complex structural and functional barrier characterized by low permeability, low pinocytosis, and lack of fenestration, thus providing a compartmental resistance across the blood and brain. It facilitates the passing of nutrients and oxygen from the blood and restricts accumulation of neurotoxins in the CNS. The BBB is constituted of different types of cells. Endothelial cells that line the brain capillaries are connected by tight junctions that interact with other supporting cells, including astrocytes, mast cells, pericytes, and microglia, to form a neuro-vascular unit [43]. It is known that, in GBM, morphological changes occur in the BBB, due to which, it shows enhanced vascular permeability and drug uptake. In pathological conditions, miRNA can pass from the brain tissue to the blood, thus serving as biomarkers for CNS diseases [44,45,46]. Moreover, extracellular vesicles from breast cancer cells, containing miR-181c have been found to disrupt the BBB and promote brain metastasis by downregulating 3-phosphoinositide-dependent protein kinase 1 (PDPK1) and altering the actin filaments [47]. Thus, miRNA-based therapeutics are attractive in GBM. One study has shown that miRNA-27a-3p mimics target the membrane protein, Aquaporin-11 (AQP11), located in the endothelial cells of brain capillaries, and has protective effects on BBB integrity in rats with intracerebral hemorrhage [48]. Thus, miRNAs have the potential to serve as potential therapeutics and biomarkers in GBM and other CNS diseases. 

### 2.3. miRNA-Based Therapeutics in Glioma

miRNA levels can be modulated with various tools either at the level of their biogenesis or by affecting their stability. These include gene therapy to sponge miRNAs, oligonucleotide therapy, and RNA delivery systems using nanoparticles [49].

#### 2.3.1. Targeting miRNA Biogenesis by Using Small Molecule Inhibitors

Hypoxia in tumors is known to downregulate Drosha and Dicer, thus reducing miRNA biogenesis and increasing tumor progression mediated through the ETS1/ELK1 transcription factors. Downregulating these factors can rescue miRNA biogenesis and reduce tumor progression in ovarian cancer [50]. Targeting oncogenic miRNAs by knock down of argonaute 2 (AGO2) has shown to induce apoptosis in myeloid leukemia [51]. However, these strategies can target the entire miRNA spectrum in the body and could lead to severe toxicities. Specific drugs that selectively inhibit certain miRNAs have been used in glioma. Phenformin, an antidiabetic drug, reduces stemness and growth in GBM by increasing the expression of the miRNAs, let-7, miR-124, and miR-137 [52]. Erismodegib, a phase III drug for medulloblastoma, induced apoptosis in glioma stem-like cells (GSCs) by suppressing the expression of miR-21 and suppressed EMT by upregulating miR-128 in GSCs [53]. Further, curcumin, a polyphenolic compound blocks the action of a lncRNA lncRNA-*ROR*, which acts as a competing endogenous RNA (ceRNA) to inhibit miR-145 activity in prostate cancer stem cells [53] and has been suggested to be used in clinical trials for the treatment of human brain tumors owing to its low toxicity [54]. Thus, small molecule inhibitors modulating miRNA biogenesis or expression have been used for combating cancer, including in glioma.

#### 2.3.2. Oligonucleotide-Mediated Therapy

miRNA mimics have been used to increase the expression of a specific miRNA particularly for the restoration of tumor suppressor miRNAs in the cell. These are synthetic double stranded (ds)RNA molecules that have the same sequence as natural miRNAs and can integrate into the RISC to perform the anti-tumorigenic functions of the missing natural miRNA [49]. Chemically modified single strand RNA (ssRNA) mimetic therapy has also shown success in the brain of Huntington disease (HD) mouse model. ssRNA, when stereo tactically injected into the brain, could bind to Ago2, and inhibits several miRNA targets without any carrier [55]. The levels of tumor suppressor miRNAs could also be increased in the brain by reactivating their transcription, by activating silenced hypermethylated miRNA promoter sites, by replenishing missing miRNA genomic regions by CRISPR/Cas9, or by inhibiting miRNA sponges such as lncRNAs and circRNAs, which form complex regulatory networks in the brain [56,57].

Anti-miRNA therapy in glioma is directed against the oncogenic miRNAs expressed in the brain. This is facilitated by antisense oligonucleotides (ASOs) that bind to the mature miRNA and block their functions. These are referred to as anti-miRs, antagomiRs, anti-miRNA oligonucleotides (AMOs), or locked nucleic acids (LNAs) [49]. Other types of ASOs include miRNA masks, which bind to miRNA-recognizing elements (MREs) on mRNA and inhibit mRNA activity; also, small RNA zippers [58], which are ss DNA LNA molecules that block miRNA by connecting mature target miRNAs, end to end through a complementary interaction and forming a DNA-RNA duplex [59].

AMOs are ss RNA molecules, ~20 nts in length, that are chemically modified using 2′-O-methoxyethyl and phosphorothioate and selectively bind to the mature miRNA to form a miRNA-RNA duplex that is sensitive to degradation by the endonuclease RNase H [44]. In one study, delivery of anti-miR-21 antisense oligodeoxynucleotides (antisense-ODN) in a GBM xenograft model led to apoptosis of tumor cells [60]. AntagomiRs are cholesterol-conjugated ssRNA designed to inhibit miRNA molecules in vivo. Delivery of antagomiR-27a in U87 GBM tumor xenografts in mice resulted in reduced proliferation and invasiveness [61,62]. LNAs are anti-miRs which have a more rigid conformation due to the presence of a methylene bridge between the 2′-O and 4′-C atoms of their ribose ring, and are thus more nuclease resistant and have enhanced binding to their target miRNA [44]. Delivery of anti-miR-21 LNA sequences using RNA nanoparticles (RNP) reduced the expression of oncogenic miR-21 and resulted in tumor regression in GBM [63]. miRNA sponges are another mode of miRNA modulation, wherein artificially synthesized ncRNAs, lncRNA, or circRNAs with multiple MREs are delivered into the cell to sponge and modulate miRNA function. In one study, a lentiviral-based miRNA sponge targeting miR-23b delivered in GBM cells and orthotopic mouse model could reduce tumor malignancy [64].

#### 2.3.3. RNA Delivery Systems

The complexity of the CNS and other physiological barriers pose challenges to the delivery of drugs and RNA-based therapeutics in neurological disorders and cancers such as glioma. Thus, efficient, biocompatible carrier systems are required for delivery of nucleic acid loads to a precise location in the brain without being subjected to degradation. The nanotechnology-based carriers, metallic nanoparticles, polymeric systems, liposomes, and lipid-based carriers have shown efficiency in delivering miRNA therapeutics in glioma [36]. Spherical gold (Au) nanoparticles have a densely packed surface with oligonucleotides through a thiolate-Au interaction. These nanoparticles have low toxicity, have high transfection efficiency, and can cross the BBB. However, the RNA loaded onto the surface are chemically modified and can also elicit immune reactions due to constant exposure [49]. As discussed above, delivery of LNA against miR-21 via multi-valent folate (FA)-conjugated three-way-junction (3WJ)-based RNA nanoparticles (RNP) specifically targeted to miR-21, rescued tumor suppressors including PTEN and program cell death 4 (PDCD4), and induce apoptosis in GBM cells in vitro and in vivo [63]. Moreover, delivery of anti-miR-21 ASOs using lipid-polymer nanoparticles (LPNs) has shown efficient targeting in GBM [65]. Cell-penetrating peptides, in which cationic peptides are linked to nucleic acids, are another efficient form of delivery. The delivery of an anti-miR 221 peptide-nucleic acid (PNA) along with temozolomide (TMZ), mesoporous silica nanoparticles (MSNPs), could induce apoptosis in TMZ-resistant GBM cells [66]. The first-in-human phase I clinical trial was conducted for MRX34, a liposomal miR-34a in patients with advanced refractory solid tumors, and showed signs of antitumor activity in a subset of patients [67]. Thus, strategies that allow efficient internalization and delivery of nucleic acids into specific tumor regions can been used capably for miRNA-based therapeutics in GBM.

### 2.4. miRNAs as Biomarkers in Glioma

A biomarker is a biological indicator that can be measured objectively to understand the presence of prognosis of a disease. The importance of biomarkers in brain tumor diagnostics has grown over the past decades, as a liquid biopsy from blood or the cerebrospinal fluid (CSF) would spare surgical intervention. Though obtaining CSF is more invasive, it is more useful due to enrichment of miRNAs in the CSF [44].

miR-21 has been pointed out to be the most powerful miRNA in brain cancer diagnostics, as per a meta-analysis from 2015 [68]. In one study, levels of miR-21 expression could differentiate between glioma patients and healthy controls, although not between glioma and other brain tumors [69]. In another study, circulating miRNA biomarkers were used for the diagnosis of glioma. Combined expression analysis could distinguish between glioma patients and healthy controls, and miR-16 expression levels could differentiate between GBM (grade IV) and lower grades of gliomas [70]. Moreover, the expression levels of miR-222 and miR-124-3p along with miR-21 in the serum have been shown to help in the differential diagnosis of gliomas at onset, in the grading of gliomas, and in assessing post-surgical recovery [71].

miRNAs that are secreted into the circulation either circulate freely or are packaged into exosomes, in which these are protected from ribonuclease-mediated degradation by the lipid-bilayer of the exosomes [72]. One study has shown that miR-21 levels in extracellular vesicles (EVs) derived from CSF of GBM patients was around 10 times higher than that from normal controls, thus highlighting its importance as potential biomarker for GBM [73]. Similarly, Yang et al. showed that miR-221 levels in serum exosomes was higher in GBM patients, and its levels increased in GBM (grade IV) comparing to lower grade gliomas [74]. Further, exosomal miRNAs could also serve as independent prognostic markers in glioma. miR-301a levels were found to be associated with a longer overall survival (OS) [75]. Thus, exosomal miRNAs in body fluids have enormous potential for biomarker development in the diagnosis and prognosis of glioma.

## 3. Long Non-Coding RNAs

In the past several years, lncRNAs have been found to affect gene expression in various biological and physiopathological conditions involving neurodegeneration, cellular stress response, and cancer [76,77,78,79]. These RNA molecules which are >200 nt in length, generally lack protein-coding capacity and mostly contain the 5′m7G caps and 3′ poly(A) tails structures like mRNAs; however, they are more dynamic and tissue-specific compared to mRNAs, thus suggesting they play multiple functional roles [80]. 

### 3.1. Classification of LncRNAs

LncRNAs can be characterized based on their genomic location, subcellular localization, or presence of unique structures. Based on their genomic location relative to protein-coding genes, these are characterized as sense, antisense, bidirectional, intronic, or intergenic (linc) RNAs. Antisense (AS) lncRNAs are transcribed by the AS RNA strand of the protein-coding gene while sense lncRNAs overlap with exons or introns of different protein-coding genes in the sense RNA direction. Bidirectional lncRNAs are transcribed from the promoter of non-coding genes in the opposite direction, intronic lncRNAs are transcribed completely from introns whereas intergenic lncRNAs are gnomically located between two protein-coding genes [76,81]. Furthermore, lncRNAs are also classified as nuclear or cytoplasmic based on their localization, which also determines their functions. Nuclear lncRNAs are involved in gene regulation through transcriptional regulation or chromatin remodeling, while cytoplasmic lncRNAs function in processes such as maintaining mRNA processing, stability, or protein regulation [82,83,84].

### 3.2. Functional Mechanisms of LncRNAs in Glioma

LncRNAs are involved in glioma initiation, progression, and infiltration/invasion processes [80,85,86]. These molecules are implicated in glioma progression by regulating the cellular proliferation, apoptosis, differentiation, stem-cell self-renewal, and response to hypoxic stress. At the molecular level, lncRNAs regulate glioma progression by epigenetic regulation, transcriptional and post-transcriptional modulation, and translational regulation [87]. An extensive list of recent studies on oncogenic and tumor suppressive lncRNAs associated with glioma and their underlying functional mechanisms has been provided in Table 1.

#### 3.2.1. Post-Transcriptional Regulation of LncRNAs as a miRNA-Sponge in Glioma Oncogenesis and Therapy Resistance

Many of the glioma-associated lncRNAs function as miRNA sponges, to diminish the availability of the transcribed miRNAs, resulting in elevation of their downstream targeting genes [88]. LncRNAs function as ceRNA to regulate the miRNA/mRNA axis in various human diseases [89]. One such lncRNA *OXCT1* antisense RNA 1 (*OXCT1-AS1*) is found to be upregulated in GBM patients, functions as a ceRNA of miR-195 to promote the upregulation of CDC25A expression, and in turn, glioma progression [90]. Other examples of the ceRNA-miRNA-mRNA axis in glioma include the lncRNA *HOXA-AS3*/miR-455-5p/USP3 axis, which promotes malignancy in GBM, and the *Linc00152*/miR-103a-3p/FEZF1 axis known to enhance malignancy in glioma stem cells (GSCs) [91,92]. Moreover, *LINC00115* enhances GSC self-renewal by competitively binding to miR-200s, thereby enhancing ZEB1 signaling, GSC self-renewal, and tumorigenicity [93]. *NEAT1* lncRNA is upregulated in glioma to inhibit the expression of miR-132 by removing miR-132′s negative regulation on SOX2, thus promoting gliomagenesis [94]. SNHG12 is an oncogenic lncRNA involved in the acquired resistance to TMZ therapy in GBM. Loss of promoter methylation led to high expression of *SNHG12* by the transcription factor SP1. *SNHG12* further served as a miRNA sponge in the cytoplasm leading to activation of MAPK1 and E2F7 and conferring TMZ resistance to GBM cells [95]. In another study, *SNHG12* functions in GBM by inhibiting the tumor suppressor, miR-627-5p, resulting in the activation of two oncogenic targets, CDK6 and SOX-2. Treatment with anti-CDK6 inhibitor palbociclib in TMZ-resistant GBM PDX mouse model led to anti-tumorigenic phenotypes including reduced SNHG12 transcript levels [96]. Another lncRNA lnc-*TALC* (temozolomide-associated lncRNA in glioblastoma recurrence) found to be upregulated in TMZ-resistant GBM cells, would competitively bind miR-20b-3p to increase c-Met expression; suggesting the contribution of lnc-*TALC* contributes to TMZ resistance and GBM recurrence in clinical patients [97].

#### 3.2.2. LncRNAs Regulate Gliomagenesis through Protein Anchoring and Modulation

LncRNAs bind directly to proteins to regulate their signaling or binding with interacting partners, thus regulating their function. A lncRNA SWI/SNF complex antagonist associated with prostate cancer 1 (*SChLAP1*), which is known to promote prostate cancer has now been shown to have role in the development of GBM. High level of *SChLAP1* in primary GBM tumors binds to the heterogeneous nuclear ribonucleoprotein L (HNRNPL), which led to enhanced binding between HNRNPL and α-actinin-4 (ACTN4), and suppression of ACTN4 degradation. This would, in turn, lead to enhanced nuclear factor kB (NF-κB)-signaling activity, which is associated with cancer development [98]. *NEAT1* lncRNA is induced by the EGFR pathway in glioma which then sequesters the chromosome modification enzyme EZH2 to facilitate repression of certain target genes and to promote nuclear translocation of β-catenin, thus activating the WNT/β-catenin oncogenic pathway in GBM [99]. Binding of another lncRNA *HOTAIR* to EZH2 of the polycomb repressive complex (PRC) 2 leads to transcriptional silencing of tumor suppressor genes in glioma. Treatment with AQB (AC1Q3QWB), which is a HOTAIR-EZH2 inhibitor, blocks PRC2 recruitment in PDX models of GBM, thus combating the tumor phenotype [100].

#### 3.2.3. Transcriptional Regulation by LncRNAs in Glioma

Dysregulated lncRNAs in glioma have been found to alter the promoter activity of oncogenic genes by facilitating their interactions with transcription factors. LncRNA PAX-interacting protein 1- antisense RNA1 (lncRNA *PAXIP1-AS1*) is overexpressed in glioma and is involved in glioma cell invasion and angiogenesis. *PAXIP1-AS1* could enhance the promoter activity of kinesin family member 14 (KIF14) involved in cell cycle and mitotic progression by recruiting the transcription factor ETS1 [101,102]. Another lncRNA called PRC2 and dead-box helicase 5 (DDX5) associated lncRNA (*PRADX*) could enhance tumorigenesis in GBM by suppressing the expression of the UBX domain protein 1 (UBXN1) protein which is a negative regulator of NF-κB signaling, thus enhancing NF-κB signaling leading to GBM cell proliferation. *PRADX* does so by increasing trimethylation of H3K27 in the UBXN1 gene promoter by recruiting the PRC2/DDX5 complex, thus transcriptionally regulating GBM tumorigenesis [103].

Epi-transcriptomic changes lead to lncRNA regulation in GBM. The m^6^A demethylase AlkB homolog H5 (ALKBH5), which is induced under hypoxic conditions in GBM erases the m^6^A deposition from the lncRNA *NEAT1*, thus stabilizing *NEAT1* and enhancing the formation of *NEAT1*-associated paraspeckle assembly. These paraspeckles induce the relocation of the transcriptional repressor splicing factor proline and glutamine rich (SFPQ) to the paraspeckles from the chemokine C-X-C motif chemokine ligand 8 (CXCL8) promoter leading to upregulation of CXCL8/interleukin-8 (IL-8) expression. Expression and secretion of CXCL8/IL-8 results in an immunosuppressive tumor microenvironment (TME) by recruitment of tumor-associated macrophages (TAM) leading to tumor progression [104].

#### 3.2.4. Role of LncRNAs in Epigenetic Regulation in Glioma

LncRNAs regulate gene expression through epigenetic regulation in glioma by recruiting chromatin modifiers to a particular genomic location, as a scaffold, thus keeping away chromatin modifiers from their regulatory targets as decoys, and also act as modulators of the 3D organization of genomes [80]. The lncRNA HOTAIRM1 regulates the expression of the HOXA1 gene by sequestering the epigenetic modifiers and demethylases G9a and EZH2 away from the transcription start site of HOXA1 gene, thus mediating demethylation of histone H3K9 and H3K27, and thus, reducing the DNA methylation of the HOZA1 gene, resulting in GBM cell proliferation, migration, invasion, and inhibiting apoptosis [105]. Furthermore, the lncRNA SNHG6 (small nucleolar RNA host gene 6) stabilized by its binding partner, NCBP3 (Nuclear cap-binding subunit 3), inhibits GBX2 (gastrulation brain homeobox 2) by mediating H3K27me3 modification induced by PRC2 (polycomb repressive complex 2) to facilitate the malignant progression in glioma [106]. In another study, lncRNA ZFAT-AS1, promotes the transcription of CDX2 (caudal type homeobox 2) by mediating histone H3 methylation on lysine 27 in a PRC2 dependent manner to facilitate the malignant biological behavior of glioma cells [107].

### 3.3. Exosomal LncRNAs in Glioma

Exosomes are a means of intercellular communication in the tumor microenvironment that transport lipids, nucleic acids, proteins, and bioactive compounds between cells, and participate in tumor initiation and progression by creating a tumor-permissive microenvironment. Exosomes can cross the BBB and are found to be present in almost all body fluids, thus rendering them as attractive biocompatible molecules for delivering therapeutic targets and serving as biomarkers in glioma [72]. Different types of ncRNAs including lncRNA, circRNA, miRNA, piRNA, and tsRNA are packaged into the double-layered exosomes, and are thus protected from the extracellular proteases and nucleases, and are considered attractive therapeutic targets in cancers including glioma.

Chronic inflammation in the brain due to the presence of glioma cells lead to infiltration of reactive astrocytes, which pose an invasive phenotype at the interface with glioma cells [108]. One of the underlying mechanisms by which glioma cells lead to activation of these astrocytes is by the secretion of exosomes. Glioma-derived exosomes shuttle a lncRNA called lncRNA activated by TGF-β (lncRNA-*ATB*) to astrocytes that activate astrocytes through the suppression of miR-204-3p in an Ago-dependent manner, and activated astrocytes could, in turn, promote the migration and invasion of glioma cells [108]. Another lncRNA antisense transcript of hypoxia-inducible factor-1α (*AHIF*) is found to be upregulated in cancerous tissues and exosomes derived from *AHIF*-knockdown GBM cells inhibited cell growth and invasiveness in GBM cells through regulation of factors associated with angiogenesis and migration in exosomes [109]. Further, *Linc01060*-containing exosomes derived from hypoxic GSCs (H-GSCs) drive disease progression in glioma. *Linc01060* transferred to glioma cells through exosomes binds to and stabilizes the transcription factor myeloid zinc finger 1 (MZF1), which then translocate to the nucleus and promotes *c-Myc* transcriptional activities including accumulation of HIF1α, which binds to Linc01060 promoter and upregulates the lncRNA, leading to glioma progression [110]. Another study reported that high amounts of lncRNA SBF2 antisense RNA 1 (lncRNA *SBF2-AS1*) were present in exosomes derived from TMZ-resistant GBM cells and it could lead to TMZ resistance in chemo-responsive GBM cells. *SBF2-AS1* acts as a ceRNA for miR-151a-3p leading to activation of the miRNA’s target, X-ray repair cross complementing 4 (XRCC4) which repairs double stranded breaks (DSB) caused by TMZ-induced DNA damage, thus resulting in TMZ resistance [111]. In a study by Ma et al., it was shown that the *HOTAIR* lncRNA-containing exosomes are secreted by glioma cells to endothelial cells where the lncRNA increases angiogenesis by regulating vascular endothelial growth factor (VEGF)-A levels [112]. Similarly, linc-*CCAT2* could be transferred to endothelial cells to increase angiogenesis by activating VEGF-A and TGFβ [113].

### 3.4. LncRNAs as Potential Biomarkers in Glioma

Glioblastoma microvesicles or exosomes that are secreted by GBM tumor cells have been shown to be clinically relevant biomarkers in the serum of GBM patients [114], providing diagnostic information through a blood test (liquid biopsy) in cancer patients. The tissue-specific expression of lncRNAs in different cancer types and their correlation with tumor-grade makes them attractive prognostic and biomarker targets. For example, the association of *HOXA11-AS* with the mesenchymal subtype of GBM helps in profiling of glioma subtypes [115]. Additionally, data analysis shows that the lncRNA *HOXA-AS3* is associated with tumor grade and shows a poor prognosis in glioma [116], suggesting the potential of lncRNAs as biomarkers in cancer diagnostics.

### 3.5. Therapeutic Targeting of LncRNAs in Glioma

Antisense oligonucleotide (ASO) therapy to correct aberrant lncRNA functions in cancers is upcoming and promising [117]. Short (21–28 nt) chemically stabilized ASOs targeting the binding sequence of lncRNA to its binding partner have been shown to alter dysregulated lncRNA functions preclinically. A 2′-deoxy-2′-fluoro-D-arabinonucleic acid (2′-FANA) modification protects oligonucleotides from an RNase H-mediated degradation [118]. In a study in melanoma, it was shown that 2′-FANA modified oligonucleotides either antisense to lncRNA *SLNCR1* androgen receptor (AR)-binding sequence or mimicking the AR binding site of *SLNCR1*, inhibits the interaction between the two, suppressing melanoma invasion [119]. In glioma, ASOs targeting the lncRNA taurine up-regulated gene 1 (*TUG1)* coupled with drug delivery led to GSC differentiation and repressed cell growth in vivo [120]. In a preclinical study, matastasis associated lung adenocarcinoma transcript 1 (*Malat1*)-specific ASO along with nucleus-targeting peptide gold nanoparticles, reduced *Malat1* expression and suppressed metastatic tumor nodule formation in vivo [121]. Moreover, lncRNA *Malat1* has been targeted using ASOs in several cancer types preclinically to reduce tumor growth in vitro and in vivo [121,122].

**Table 1 biomedicines-10-02031-t001:** Non-coding RNAs associated with Glioma.

Type of ncRNA	Name	Functional Mechanism	Reference
Oncogenic miRNAs	miR-10b	Essential for viability of GBM cells; controls cell proliferation, survival, migration, invasion, and epithelial-to-mesenchymal transition	[123]
	miR-1246	Drives the differentiation and activation of myeloid-derived suppressor cells	[124]
	miR-30e*	Promotes invasiveness by disrupting the NF-κB/IκBα loop	[125]
	miR-637	Drives the malignant phenotype of glioma by suppressing HMGA1 in vitro and in vivo	[126]
	miR-30b-3p	EV-packaged miR-30b-3p (EV-miR-30b-3p) directly targeted ras homolog family member B (RHOB), increases TMZ resistance	[127]
	miR-155 and the miR-23a cluster	Down regulate the tumor suppressor MAX interactor 1 (MXI1)	[128]
Tumor suppressive miRNAs	miR-3189	Controls cancer cell metabolism by inhibiting GLUT3 expression	[129]
	miR-124	miR-124-extracellular vesicles (EVs) inhibits M2 microglial polarization	[130]
	miR-593	Regulates the Eph receptor tyrosine kinase Eph receptor B2 (EphB2) levels	[131]
	miR-205	ERBB3 overexpression is sustained by epigenetic inactivation of the oncosuppressor miR-205	[132]
	miR-185-5p	Regulates the expression of tumor promoting HOXB5 expression	[133]
	miR-195	Regulates expression of the cell division cycle 25A (CDC25A) phosphatase, a key regulator of cell cycle progression	[123]
	miR-198	Reduces cellular MGMT levels through binding to the 3′-UTR of MGMT to enhance TMZ sensitivity	[134]
	miR-432-5p	Regulates expression of RNA editing enzyme adenosine deaminases acting on RNA (ADAR)	[135]
	miR-375	Selective exosome exclusion of miR-375 by glioma cells inhibited the CTGF-epidermal growth factor receptor (EGFR) signaling pathway	[136]
	miR-342-3p	Regulates the expression of the transcription factor ISL2 which transcriptionally regulated VEGFA expression in GSCs	[137]
	miR-137	Has roles in cell proliferation, apoptosis, invasion, and angiogenesis and in glioma treatment	[138]
	miR-199a-3p	Regulates expression of the transcription factor EGR1 (early growth response 1) involved in GSC proliferation and self-renewal	[139]
Oncogenic lncRNAs	ALKBH5	Facilitates IL8 Secretion to Generate an Immunosuppressive TME	[104]
	TMEM44-AS1	Sequentially activated Myc and EGR1/IL-6 signaling	[140]
	HOXA10-AS	Regulates contact inhibition, cell proliferation, invasion, Hippo signaling, and mitotic and neuro-developmental pathways	[141]
	RBM5-AS1	Interacted with and stabilized sirtuin 6 (SIRT6) protein	[142]
	lnc-HLX-2-7	Modulates oxidative phosphorylation, mitochondrial dysfunction, and sirtuin signaling pathways	[143]
	OXCT1-AS	Acts as a ceRNA of miR-195 to enhance CDC25A expression	[90]
	lnc PRADX	Suppressed UBXN1 expression through promoter methylation and promoted NF-κB activity	[103]
	Linc 01060	Directly interacts with the transcription factor myeloid zinc finger 1 (MZF1), promotes its nuclear translocation and MZF1-mediated c-Myc transcriptional activities	[110]
	LPP-AS2	Functions as a ceRNA to regulate EGFR expression by sponging miR-7-5p	[144]
	SNHG12	Acts as s sponge for miR-129-5p, leading to upregulation of MAPK1 and E2F7	[95]
	HOTAIR	GBM-serum-EV-enclosed HOTAIR competitively binds to miR-526b-3p to increase EVA1 expression	[145]
	MIR22HG	Induces the Wnt/β-catenin signaling pathway	[146]
	PAXIP1-AS1	Upregulates the KIF14 promoter activity by recruiting transcription factor ETS1	[102]
	LINC00115	Upregulates ZEB1 and promotes ZNF596 transcription by acting as a miRNA sponge for miR-200s	[93]
	SChLAP1	Binds to heterogeneous nuclear ribonucleoprotein L (HNRNPL) to stabilize actinin alpha 4 (ACTN4) through suppression of proteasomal degradation	[98]
	MALAT1	Regulated by NF-κB and p53. MALAT1 is a potential target for the chemo sensitization of GBM	[147]
	lnc-TALC	Competitively binds miR-20b-3p to facilitate c-Met expression	[97]
Tumor suppressive lncRNAs	ARST	Inhibits ALDOA-mediated actin cytoskeleton integrity	[148]
	lncRNA MEG3	Regulating cell adhesion, EMT, and cell proliferation	[149]
Oncogenic circRNAs	circNEIL3	Drives macrophage infiltration and immunosuppression by stabilizing IGF2BP3 (insulin-like growth factor 2 mRNA binding protein 3)	[150]
	circATP5B	Acts as miR-185-5p sponge to upregulate HOXB5 expression	[133]
	ASAP1	Increases the expression of NRAS via sponging miR-502-5p.	[151]
	circ E Cadherin	Stimulates EGFR signaling independent of EGF	[152]
	circ SMO	Encodes SMO-193a.a., required for sonic hedgehog (Shh) induced G protein-coupled-like receptor smoothened (SMO) activation	[153]
	circ ATXN1	Targeted miR-526b-3p to upregulate MMP2/VEGFA expression	[154]
	circ FOXO3	Sponges both miR-138-5p and miR-432-5p to increase expression of nuclear factor of activated T cells 5 (NFAT5)	[155]
	circPTN	Sponges miR-145-5p and miR-330-5p to rescue downregulation of SOX9/ITGA5 in glioma cells	[156]
	circ 002136	Sponges miR-138-5p to enhance SOX13 mediated angiogenesis	[157]
	circ DICER1	Acts as a molecular sponge to adsorb miR-103a-3p/miR-382-5p and enhance ZIC4 in glioma-exposed endothelial cells (GECs)	[158]
	circ NT5E	Sponges miR-422a to inhibit the miRNA’s activity	[159]
Tumor suppressive circRNA	CDR1as	Stabilizes p53 protein by preventing it from ubiquitination by MDM2	[160]
	circ MAPK4	Modulates miR-125a-3p to enhance the activity of the p38/MAPK signaling pathway	[161]
	circ LINC-PINT	Encodes a 87-aa peptide that interacts with polymerase associated factor complex (PAF1c) and inhibits the transcriptional elongation of multiple oncogenes	[162]
	circ FBXW7	Antagonizing USP28-induced c-Myc stabilization, thus reducing the half-life of c-Myc	[163]

## 4. Circular RNAs 

circRNAs are single-stranded ncRNA molecules with a covalently closed loop structure and lack the 5′-cap and 3′-polyA tail structures [164]. They are discovered to be present in large numbers in the brain and in other tissue types including the lung, liver, heart, stomach, kidney, and colon, with an enrichment in neuronal tissues. CircRNAs were initially considered as ‘splicing noise,’ but with the advent of high throughput RNA sequencing and circRNA detection algorithms, large numbers of circRNAs have been detected in normal as well as cancer tissues. Due to their presence in the serum and other body fluids, circRNAs hold high prognostic and therapeutic value in cancer and their high levels of expression in brain tissues make them an attractive target in glioma [165]. Please refer to Table 1 for a comprehensive list of circRNAs associated with glioma.

### 4.1. Biogenesis and Classification of CircRNAs

CircRNAs are primarily synthesized by back splicing of pre-mRNAs wherein a downstream 5′-donor splice site is connected to an upstream 3′-acceptor splice site [164,166,167]. The vast majority of circRNAs (>80%) are synthesized in the cytoplasm mostly from the exon of protein-coding genes, while single gene loci can also generate circularization patterns [167,168,169].

CircRNAs are classified based on their biogenesis from different genomic locations [164]. Exonic circRNA (ecircRNAs) are derived from one or more exons through alternative splicing and are localized in the cytoplasm [164]. ecircRNAs are formed either by direct base pairing of flanking introns mediated by reverse complementary *Alu* sequences, or by lariat formation of flanking introns that facilitates back splicing and circularization of exons, or through binding of RNA binding proteins (RBPs)/splicing factors, to specific sequence motifs in flanking introns. Intronic circRNAs are generated by presence of 7nt GU-rich elements at the 5′-splice site and an 11nt C-rich sequence at the branch point site that is sensitive to debranching enzymes and are localized in the nucleus [164,170]. Intergenic lncRNAs are generated from the region in-between genes. Additionally, circRNAs are generated from certain lncRNAs that contain short open reading frames (ORFs) such as the lncRNA *LINC-PINT* [171].

### 4.2. Expression of CircRNAs in Glioma

The enrichment of circRNAs in neuronal tissues suggests their association with various brain disorders, including cancer. Furthermore, mounting evidence of circRNA expression in glioma emphasizes their function in glioma development and tumorigenesis. In an RNA sequencing differential expression analysis using Illumina Hi-Seq, between five GBM tumors and five normal brain tissues, a total of 1411 differentially expressed circRNAs were analyzed, with 206 upregulated and 1205 downregulated circRNAs [172]. In another study to find differentially expressed circRNAs in glioma tissue, the authors used five different methods to screen for circRNAs between three glioma tissues and paired normal tissues—CIRCexplorer2, circRNA-finder, CIRI, find-circ and MapSplice2 and discovered twelve differentially expressed circRNAs in glioma tissues [173]. In a recent study, N Vo et al., have used the technique of exome capture RNA sequencing to detect and characterize circRNAs across >2000 cancer samples including glioma. Using capture, the authors have built a comprehensive catalog of circRNAs directly detected in cancer tissues—called MiOncoCirc. They have also identified a novel class of circRNAs called read-through circRNAs, which contained exons originating from different genes [174].

### 4.3. Functions of CircRNAs in Glioma

#### 4.3.1. CircRNAs Act as a miRNA Sponge

circATP5B acts a miRNA sponge and promotes proliferation of GSCs. circATP5B sponges miR-185-5p to upregulate homeobox 5 (HOXB5) protein, which regulates transcriptional increase of IL-6 and promotes GSC proliferation through the JAK2/STAT3 signaling pathway [133]. Further, in recurrent GBM tissues, (circ)RNA ADP-ribosylation factor GTPase activating proteins with Src homology 3 domain, ankyrin repeat and Pleckstrin homology domain 1 (circASAP1) was found to be upregulated and conferred TMZ resistance in GBM cells. circASAP1 is a miRNA sponge for miR-502-5p and enhanced the expression of oncogene NRAS, thus causing therapy resistance in GBM [151]. circMMP9 targeted miR-124 and increased GBM migration and invasion through activation of the cyclin-dependent kinase 4 (CDK4) and aurora kinase A (AURKA) signaling axis [175].

#### 4.3.2. Role of CircRNAs and Its Encoded Peptides in Glioma Development

Through circRNA sequencing analysis in glioma and normal tissues, circNEIL3 was identified to be involved in glioma development in vitro and in vivo. circNEIL3, which was cyclized by the EWS RNA-binding protein 1 (EWSR1) RNA binding protein, stabilizes the oncogenic protein insulin-like growth factor 2 mRNA binding protein 3 (IGF2BP3) by preventing its degradation mediated by HECTD4 ubiquitin ligase [150]. 

Another study has identified a circular EGFR RNA called circ-EGFR, which is aberrantly activated in more than 50% of adult GBM cases. Circ-EGFR encodes a novel EGFR variant termed rolling-translated EGFR (rtEGFR) which interacts with EGFR, maintains its membrane localization, and thus leads to enhanced downstream signaling and cancer progression. circ-EGFR correlated with poor prognosis in GBM patients and repression of rtEGFR in GSCs inhibited GBM tumorigenesis [176]. The EGFR pathway activity in glioma is regulated by another circRNA circular E-cadherin (circ-E-Cad) RNA, which encodes a secretory E-cadherin protein variant (C-E-Cad) synthesized through multiple-round open reading frame translation. Inhibition of C-E-Cad enhances the anti-tumor effect of anti-EGFR therapies [152]. Similarly, a variant of the Hedgehog signaling component, G protein-coupled-like receptor smoothened (SMO), called SMO-193a.a was encoded by the circRNA circ-SMO. Expression of SMO-193a.a that is regulated by a Shh/Gli1/FUS/SMO-193a.a. signaling axis, correlates with poor prognosis in GBM patients [153]. 

CircRNAs with a tumor suppressive role have also been identified in GBM. Circ-FBXW7 was identified through circRNA deep sequencing using ten GBM patient tissues and paired normal tissues. Circ-FBXW7-encoded protein FBXW7-185aa had low levels in GBM samples and its overexpression could inhibit proliferation and altered cell cycle in GBM cells [163].

#### 4.3.3. Exosomal CircRNAs Serve as Potential Biomarkers

CircRNAs are found to be enriched and stable in exosomes (exo-circRNA) [177]. Due to the enrichment of exosomes in CSF and serum, exo-circRNA can be used as potential biomarkers for GBM prognosis. In one study, differential circRNA expression was analyzed from EVs isolated from glioma U251 cells (Nor-EVs) and EVs isolated from radiation-resistant RR-U251 cells (RR-EVs). Through RNA-seq, a total of 1235 circRNAs were detected, 63 upregulated and 48 downregulated in RR-U251 compared to Nor-EVs. A highly expressed candidate circATP8B4 in RR-U251 could be transferred to normal U251 glioma cells and cause radioresistance by acting as miRNA sponge [178]. Overexpression of circNEIL3 in glioma cells lead to TAM infiltration into the TME. CircNEIL3 is packaged into the exosomes by hnRNPA2B1 and is delivered to TAMs from glioma cells, afflicting an immunosuppressive phenotype by stabilizing the IGF2BP3 protein, thereby promoting glioma progression [150]. Thus, exo-circRNAs modulate the glioma TME and serve as attractive biomarkers for GBM prognosis and diagnostic purposes.

## 5. PIWI-Interacting RNAs (piRNAs)

piRNAs are animal-specific small (26–32 nt) ncRNAs which are derived from long intergenic transcripts, ncRNAs, and 3′ UTRs of protein coding RNAs [179,180]. piRNAs have conserved function in the protection of the germline tissues from destabilizing transposable elements [181]. piRNAs function in sequence-specific gene regulation and are present in the human genome in a large number (>30,000 piRNAs) with more piRNAs being identified with the advance in sequencing technologies [182]. These ncRNAs form ribonucleoprotein complexes with PIWI proteins to recruit chromatin modifiers to targets of transposable elements creating heritable epigenetic modifications, and piRNAs have also been found to function in miRNA silencing [182]. piRNAs regulate diverse functional processes including epigenetic reprogramming, transcription, translation, development, and regulation of mRNA stability [183,184].

### 5.1. piRNA Biogenesis

piRNA biogenies involves two major mechanisms, namely primary and secondary biogenesis [185]. pre-piRNAs are formed by 3′-5′ movement of RNA polymerase on the DNA heterochromatic strand. The pre-piRNA is exported out of the nucleus where it bounds to Yb bodies located around the mitochondria and first discovered in Drosophila. Yb bodies allow the interaction of PIWI proteins and piRNAs to form the piRISC by recognition of the 5′ end of pre-piRNAs [186,187].

### 5.2. piRNA Functional Mechanisms in Glioma

piRNAs were believed to be restricted to the germline for a long time. However, recent evidence suggests expression of piRNAs in somatic tissues and their dysregulated expression has been observed in 8 different cancer types [188]. Gene expression analysis of tissues other than germline from TCGA has revealed that piRNAs are expressed in both normal and malignant tissues of all 11 distinct anatomical cancer tissues and their expression in cancer tissues is found to be clinically relevant and tumor specific [189].

piRNAs have been studied to have functional role through epigenetic mechanisms in different cancers including glioma [190]. One study has found through array-based piRNA expression profiling, the expression of ~350 piRNAs in both normal and cancer tissues and a subset of piRNAs were dysregulated in cancer. Overexpression or pretreatment of a piRNA piR-8041 restores tumor suppressive properties and reduced GBM tumorigenesis in GBM cell line and mouse xenograft tumors [182]. Moreover, the piRNA interacting protein Piwil1 is enriched in GSCs and maintains GSC self-renewal and Piwil1 knockdown in GBM animal model suppresses tumor growth and promotes mouse survival [191]. Further, a study has found that the PIWIL1/piRNA-DQ593109 (piR-DQ593109) complex is a major regulator of blood-tumor barrier (BTB) permeability in glioma through the *MEG3*/miR-330-5p/RUNX3 axis, and their inhibitions could enhance efficient delivery of anti-glioma drugs to glioma tissues [192]. 

## 6. Concluding Remarks

In recent years, with the availability of only a small fraction of information on transcription of the non-coding genome, remarkable success has been achieved in the field of ncRNA biology, which has helped us understand the complex regulatory network and functions of these molecules. miRNAs function in virtually every cellular process and are functional in the development, differentiation, and homeostasis of normal physiology and of disease [193]. LncRNAs function is, in cis, to regulate the function of nearby genes by modulating their transcription or chromatin dynamics, and in trans, through structural or regulatory roles to affect mRNA stability, splicing, translation, and even signaling [76]. The upcoming circRNAs are now known to be involved in neurogenesis and dysregulated in several human diseases, including neurological disorders and cancer, thus affecting the severity of disease [194].

The potential for of ncRNAs as biomarkers and therapeutics in glioma and cancer at large is enormous and holds great promise. The fact that ncRNAs use nucleotide hybridization to bind to their targets makes them easy-to-use tools as compared to small molecule inhibitors where structure-based designing is required. Instead, for ncRNAs, finding the correct binding sequence and testing the candidate therapeutics can help prepare ncRNA-based inhibitors. Also, the detection of cancer-associated ncRNAs in blood and urine could serve as excellent biomarkers and could spare cancer patients from more invasive tissue collection procedures. Although, as pointed out above, constant exposure to chemically modified RNA molecules can elicit immune reactions, which has been a challenge for RNA medicine, and more research is being pursued to combat this issue. Research to improve the oligonucleotide delivery methods and refine their chemistry is ongoing [2,3,195,196].

miRNA therapeutics have shown promising results in preclinical studies in glioma. Thus, the identification of the right targets and a better understanding of their mechanisms of action can help to build novel treatments to overcome the therapy resistance and tumor recurrence in glioma [18]. The nanotherapeutic strategy has been an attractive delivery strategy in vitro and vivo, however, it is less successful in human applications due to issues of toxicity, stability, efficacy, and targeting, and thus, demands more research. Alternatively, patient-derived exosomes are more compatible and safer clinically. Exosomal ncRNAs are selectively packaged, secreted, and transferred between cells, and can thus modulate the TME, which can be leveraged therapeutically. Also, their ability to cross the BBB and their enrichment in body fluids make them excellent diagnostic and therapeutic agents.

The understanding of the mechanisms of lncRNAs and circRNAs is still relatively low compared to that of miRNAs, and thus, is less developed clinically; however, promising results have been shown in preclinical studies. In glioma, lncRNAs have been deeply studied for their roles as ceRNA, and hence, a better understanding of their lncRNA functions is required. Additionally, since lncRNAs function by binding to other molecules, an understanding of the structure of lncRNA binding motifs would be important. Finally, circRNA research is still in its nascent stage. With the implementation of more sequencing and other research strategies, the roles of circRNAs in glioma tumorigenesis and therapy responses will be revealed. Lastly, circRNAs and piRNAs could prove to be excellent therapeutic and biomarker targets, especially in glioma research, due to their enrichment and stability in the brain.

## Figures and Tables

**Figure 1 biomedicines-10-02031-f001:**
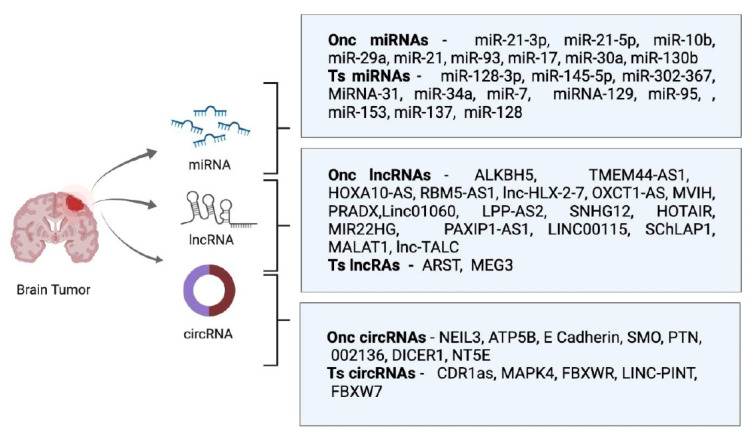
Oncogenic (Onc) and Tumor-suppressor (Ts) ncRNAs-MicroRNAs (miRNAs), long non-coding RNAs (lncRNAs) and circular RNAs (circRNAs) expressed in Glioma.

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
