# Peer review of "The Role of Non-Coding RNAs in Glioma"

_biomedicines, 2022, doi:10.3390/biomedicines10082031_

Round 1
Reviewer 1 Report
This a review paper on the functional role of non-coding RNA specifically micro RNA, long non- coding RNA and circular RNA in glioma tumorigenesis and their potential as biomarkers and therapeutics in gliomas.
There is a plethora of recent articles on non-coding RNAs including review articles. The authors should address what their review adds to the current literature and how it is different form recent reviews.
In my opinion, some relevant articles were not cited:
https://doi.org/10.1186/s12943-020-01189-3
https://doi.org/10.3389/fimmu.2022.897754
https://doi.org/10.3390/cells9112369
10.1097/RMR.0000000000000111
The article is well-written, easy to follow. The review is complete and relevant to neuro-oncology.
A few minor points to address:
1. In the introduction, the authors should specify that the gliomas with poor pronostic are the diffuse gliomas.
2. The title: Transcriptional regulation by lncRNAs in glioma is wrongly numbered.
3. The reference 20 was published in 2019
Reviewer 2 Report
In this paper, the authors focus on ncRNA expression in cancer initiation, progression, and therapy resistance, and on their role as transcriptional, post-transcriptional or epigenetic regulators. Specifically, they focus on their role in glioma tumorigenesis and therapy response.
The paper is interesting; however, in the present form, it suffers of some criticisms that should be addressed.
Major points:
1) The paragraph on lncRNAs is a list of lncRNAs with a few descriptions of their specific role. Authors should extend this section with the mechanism of action of lncRNAs in glioma.
2) The authors focus on three classes of ncRNAs (miRNAs, lncRNAs and circRNAs). However, other classes of ncRNAs have been associated to glioma development. Further ncRNAs should be mentioned (for refs see also Jacobs et al., 2018; Tamtaji et al., 2020)
3) The authors propose ncRNAs as determinants for tumor growth and invasion. Based on their interesting and innovative findings, they should propose an RNA-based therapeutic approach to counteract brain cancer. This, in line with the growing interest in the development of RNA therapeutics that should be discussed (for refs see also Damase et al., 2021; Garbo et al., 2022; Zogg et al., 2022).
Minor points:
1) Typing errors should be amended.
Round 2
Reviewer 2 Report
Accept.